# Producing aryl halides from lignin

Yongqian Liu[1,2], Yi Li[1,2], Zhiyang He[1,2], Simeng Wu[1,2], Chunhui Ma[1,2], Wei Li[1,2], Shujun Li[1,2], Zhijun Chen[1,2] ✉, Shouxin Liu[1,2] ✉ & Bing Tian[1,2] ✉

Lignin represents the most abundant biomass resource, which contains aromatic units. Lignin refinery is a promising, sustainable alternative for the production of aromatic chemicals. However, depolymerization and transformation of lignin into aryl halides, which are indispensable chemicals in both the academic and industrial communities, remain challenging. Here, we report a simple and mild method for the depolymerization and halogenation of lignin, leading to the production of useful aryl halides. Notably, hydrogen bond activation for halogenation reagents is essential for substantially increasing the reactivity, resulting a highly efficient cleavage of C–C bonds in lignin. This method is highly selective for breaking $C(sp^2)–C(sp^3)$ bonds of lignin linkages, enabling the application of precise depolymerization and halogenation reactions from lignin models to native lignin from various wood resources, which provides a sustainable and efficient access to various synthetically useful aryl halides.

Aryl halides are indispensable chemicals in both the academic and industrial communities. Owing to their unique reactivity patterns, aryl halides have played crucial roles in the development of modern chemical synthesis[1]. The carbon–halogen bonds in these chemicals have versatile and robust applications in synthetic chemistry and materials manufacture[2]. For example, aryl halides are widely used as starting materials for the synthesis of fused benzene-based advanced materials such as conjugated polymers[3–5], organic optoelectronic materials[6–8], and organic semiconductors[9–11]. Moreover, aryl halides represent the most significant precursors in cross-coupling[12–15] and nucleophilic aromatic substitution reactions or the preparation of Grignard reagents[16], which are widely used in the synthesis of pharmaceuticals, agrochemicals, and synthetic building blocks[17–20]. Currently, aryl halides are produced predominantly via the halogenation of aromatics[12,21–23], which are produced from petroleum and coal through catalytic reforming and steam cracking processes (Fig. 1a)[24–26]. However, these fossil resource-dependent routes involve complicated and multiple procedures, high energy consumptio,n and environmental pollution[27,28]. Thus, the development of efficient, environmentally friendly, and sustainable processes for chemical production from renewable biomass has attracted much attention.

Lignin, which is one of the main components of lignocellulosic biomass, represents the most abundant resource featuring aromatic structures, and is the most promising sustainable alternative for the production of functionalized aromatic compounds[29–32]. Numerous efforts have been made to valorize lignin into value-added aromatic chemicals via the depolymerization of lignin linkages under oxidative[33–37], reductive[38–40], and redox-neutral conditions[41,42]. Indeed, simple oxygen-containing aromatic products can be efficiently obtained, which can be used as useful intermediates for further diverse functionalization (Fig. 1b). Furthermore, various methodologies have been developed for the one-pot depolymerization and functionalization of lignin linkages to yield heteroatom-containing (N, S, and Si) aromatic molecules, which could increase the value of lignin valorization[43]. However, there have been no documented examples of the direct preparation of useful aryl halides from lignin, which is a fascinating and promising pathway. Owing to the complex three-dimensional structure and low reactivity of real lignin, most current methods for preparing aromatics from lignin are limited to lignin models, which are largely insufficient for practical applications with real lignin. In addition, increasing the reactivity and selectivity of halogenation is challenging due to the complex lignin structure. Thus, developing a highly efficient catalytic method, that can both depolymerize real lignin and halogenate the formed fragments to produce aryl halides is highly important and challenging.

[1]State Key Laboratory of Utilization of Woody Oil Resource, Northeast Forestry University, Harbin, China. [2]Key Laboratory of Bio-based Material Science and Technology of Ministry of Education, Northeast Forestry University, Harbin, China. ✉e-mail: chenzhijun@nefu.edu.cn; liushouxin@126.com; tianbing@nefu.edu.cn

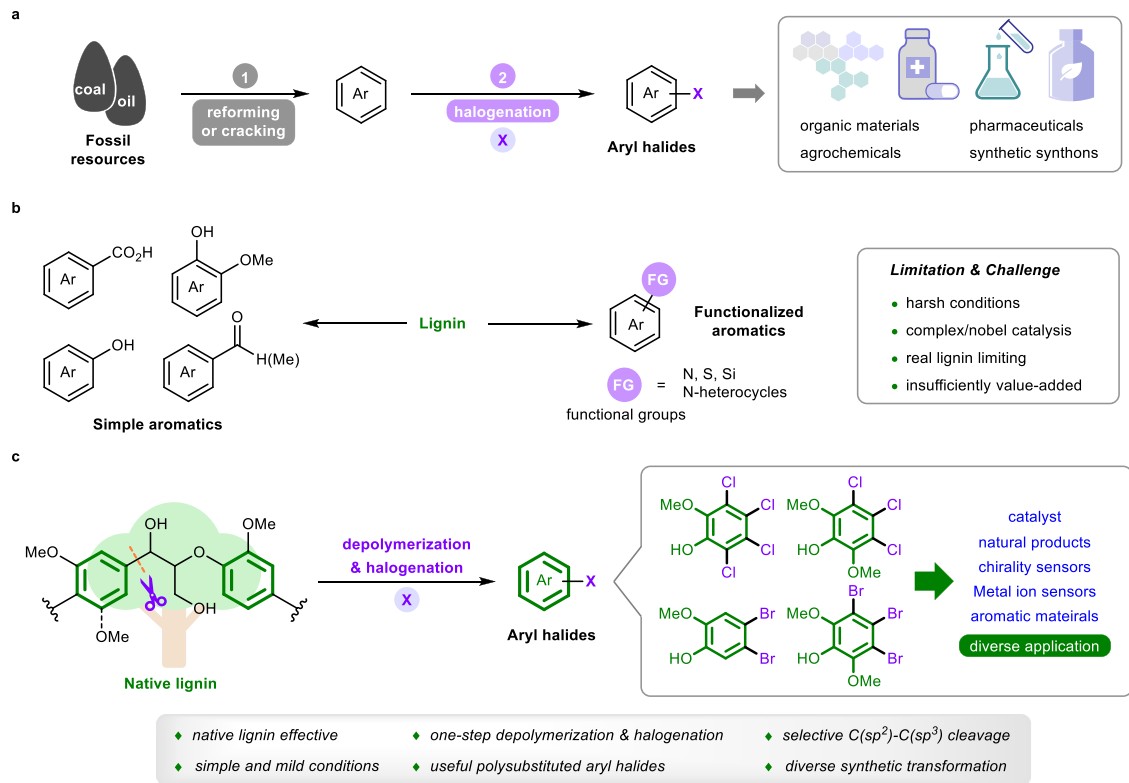

**Fig. 1 | Aryl halide production and lignin depolymerization. a** Traditional production of aryl halides. **b** Conversion of lignin into aromatic compounds. **c** This work: producing aryl halides from lignin.

Here, we describe depolymerization and halogenation of native lignin for the production of polysubstituted aryl bromides and chlorides (Fig. 1c). The reaction is initiated by the electrophilic attack of the phenyl rings of lignin by bromine or chorine cations from halogenation reagents, and the hydrogen bond activation of halogenation reagents by 2,2,2-trifluoroethanol (TFE) is key to their success. This work provides an efficient route for lignin valorization that involves the one-pot depolymerization and halogenation of native lignin under simple and mild conditions with readily/commercially available chlorine and bromine sources and selective and efficient C(sp²)–C(sp³) bond cleavage to produce useful lignin-derived aryl halides for diverse synthetic transformations.

## Results

### Screening conditions for depolymerization and halogenation

We hypothesized that the electron-rich aromatic motifs of lignin which are constructed of syringyl (S), guaiacyl (G), and *para*-hydroxy-phenyl (H) units, provide the possibility for electrophilic attack by electrophiles such as bromine and chorine cations. Subsequent bond cleavage could promote the depolymerization of lignin and afford halogenated monomers (Fig. 2a). To survey the aforementioned hypothesis, we initially employed N-bromosuccinimide (NBS), which is a commercially available and commonly used electrophilic bromination reagent, as the cut-off and functionalization tool to address lignin β-O-4 linkages (Fig. 2b). Treating lignin model **1a** with NBS at room temperature in normal solvents such as acetone, 1,4-dioxane (Fig. 2b, entries 1 and 2), ethyl acetate, and tetrahydrofuran did not disrupt the bonds, and no target products were obtained (Supplementary Table 1). This could be attributable to the inefficient electrophilicity of the Br⁺ of NBS in the reaction system, leading to unsuccessful electrophilic attack of the phenyl rings. To increase the electrophilicity of Br⁺, TFE was introduced into the reaction as a hydrogen bond activator[44]. As expected, when 0.2 mL of TFE in 1.8 mL of acetone was added, the

depolymerization successfully yielded dibrominated products **2a** and **3a** (Fig. 2b, entry 3). Increasing the dosage of TFE further promoted the reaction (Fig. 2b, entries 4 and 5). Notably, yields of 99% for **2a** and 61% for **3a** were obtained (Fig. 2b, entry 6) when TFE was used as the sole solvent, indicating efficient cleavage of the C(sp²)–C(sp³) bond with excellent selectivity. Other bromination reagents, such as N-bromophthalimide (NBP) and 1,3-dibromo-5,5-dimethylhydantoin (DBDMH), also performed well in providing brominated products with slight decreases of yields (Fig. 2b, entries 7 and 8). Moreover, dibromoisocyanuric acid (DBI) showed poor reactivity for this reaction, and only a 5% yield of **2a** was obtained (Fig. 2b, entry 9). Notably, in entries 3-9, phenol product **4a** was observed in low yields (7-16%), which was derived from the decomposition of **3a**. Compared with strong acids such as Brønsted or Lewis acids, which exhibit a strong ability to active hydrogen bonding, TFE is a mild reagent that has excellent system tolerance and is an excellent hydrogen bond donor. In addition, TFE has a good dissolving capacity due to the effect of florin atoms, which can contribute to the dissolution of the native lignin.

The depolymerization and chlorination reactions of **1a** were also investigated, and the results are shown in Fig. 2c. The chlorination reagents (1.5 equiv.) were screened using TFE as the solvent at room temperature. N-Chlorosuccinimide (NCS), 1,3-dichloro-5,5-dimethyl-hydantoin (DCDMH), and sodium dichloroisocyanurate (Na-DCC) showed poor reactivity for **1a** depolymerization and very low yields of dichlorinated product **5a** were obtained (Fig. 2c, entries 1-3). When trichloroisocyanuric acid (TCCA), which is a versatile and efficient reagent that is used in chlorination reactions due to its excellent performance for the release of active chlorine cations, was employed in the reaction, the cleavage of **1a** produced **5a** and **8a** in 48% and 32% yields, respectively (Fig. 2c, entry 4). Moreover, trichlorinated product **6a** was found in the reaction, which was probably derived from the overchlorination of primary depolymerization products. In common with bromination, phenol products **10a-12a** with different extent of

**a** Initial considerations

**b** Conditions optimization for depolymerization and bromination

| Entry | Solvent | [Br⁺] | Yield of **2a** (%) | Yield of **3a/4a** (%) |
|---|---|---|---|---|
| 1 | Acetone | NBS | 0 | 0/0 |
| 2 | 1,4-Dioxane | NBS | 0 | 0/0 |
| 3 | Acetone/TFE (9:1) | NBS | 51 | 32/10 |
| 4 | Acetone/TFE (5:1) | NBS | 63 | 36/11 |
| 5 | Acetone/TFE (1:1) | NBS | 73 | 47/14 |
| 6 | TFE | NBS | 99 | 61/15 |
| 7 | TFE | NBP | 83 | 40/16 |
| 8 | TFE | DBDMH | 77 | 33/14 |
| 9 | TFE | NDBI | 5 | trace/7 |

**c** Conditions optimization for depolymerization and chlorination

| Entry | Solvent | [Cl⁺] | Yield of **5a** (%) | Yield of **6a** (%) | Yield of **7a** (%) | Yield of **8a/9a** (%) | Yield of **10a/11a/12a** (%) |
|---|---|---|---|---|---|---|---|
| 1 | TFE | NCS | 3 | 0 | 0 | 0/0 | 0/0/0 |
| 2 | TFE | DCDMH | 2 | 0 | 0 | 0/0 | 0/0/0 |
| 3 | TFE | Na-DCC | 11 | 0 | 0 | 0/0 | 0/0/0 |
| 4 | TFE | TCCA | 48 | 22 | 0 | 32/0 | 18/0/0 |
| 5 | TFE/Methyl acetate (1:1) | TCCA | 58 | 30 | 0 | 22/0 | 9/6/0 |
| 6 | TFE/Methyl acetate (1:1) | TCCA (2.5 equiv.) | 28 | 50 | 0 | 28/0 | 16/9/0 |
| 7 | TFE | TCCA (2.5 equiv.) | 0 | 0 | 78 | 10/14 | 3/5/10 |
| 8 | TFE | TCCA (4.0 equiv.) | 0 | 0 | 92 | 0/34 | 0/0/20 |

**Fig. 2 | Initial considerations and optimization of the reaction conditions.**
**a** Initial considerations for the depolymerization and halogenation of lignin.
**b** Optimization of the conditions for depolymerization and bromination. Reaction conditions: **1a** (0.2 mmol), [Br⁺] reagent (0.9 mmol), and solvent (2 mL) at room temperature for 12 h. **c** Optimization of conditions for depolymerization and chlorination. Reaction conditions: **1a** (0.2 mmol), [Cl⁺] reagent (0.3 mmol), and solvent (2 mL) at room temperature (r.t., ~25 °C) for 12 h. **8a** 3,4-dichlorinated

product. **9a** 2,3,4,5-tetrachlorinated product. **10a** 4,5-dichloro-2-methoxyphenol. **11a** 3,4,5-trichloro-2-methoxyphenol. **12a** 2,3,4,5-tetrachloro-6-methoxyphenol. The yields of **2a** and **5a-9a** were determined by ¹H nuclear magnetic resonance (NMR) spectra with CH₂Br₂ as an internal standard, the yields of **3a**, **4a**, and **10a-12a** were detected via GC-MS analysis of the unpurified reaction mixture with *n*-octadecane and *n*-dodecane as an internal standard, respectively.

chlorination were also observed (Fig. 2c, entries 5-8), which were derived from the decomposition of **8a** or **9a**. Owing to the strong electrophilic effect of TCCA, several reaction conditions were screened to control the degree of chlorination. The use of a mixed solvent of TFE/methyl acetate (1/1, v/v) slightly increased the yield of **5a** (58%) (Fig. 2c, entry 5). Increasing the amount of TCCA to 2.5 equiv. promoted the formation of the trichlorinated product **6a** (Fig. 2c,

entry 6). When the same TCCA dosage as that used in entry 6 was used, complete chlorination occurred in the sole solvent TFE, providing the tetrachlorinated product **7a** and a mixture of di-/tetrachlorinated products **8a/9a** in 78% and 24 % yields, respectively (Fig. 2c, entry 7). A further increase in TCCA (4 equiv.) afforded the tetra-chlorinated product **7a** in a 92% yield. Thus, the depolymerization and chlorination of lignin β-O-4 models could be controlled by changing the reaction

**Fig. 3 | Reactions of lignin models.** [a]Reaction conditions: **1** (0.2 mmol), NBS (0.9 mmol), TFE (2 mL), room temperature, and 12 h. [b]Reaction conditions: **1** (0.2 mmol), TCCA (0.8 mmol), TFE (2 mL), room temperature, and 12 h. Isolated yields. **4b** 4-bromophenol. **4c**: 3-bromo-4-methoxyphenol.

conditions. Notably, both Br[+] and Cl[+] exhibited high selectivity for C(sp[2])–C(sp[3]) bond cleavage in the lignin β-O-4 models.

### Depolymerization and halogenation of lignin β-O-4 models

Under optimal reaction conditions, the scope of lignin β-O-4 models was initially investigated for depolymerization and halogenation. As the methoxyl substituent can considerably change the properties of lignin models, which can directly affect the halogenation process, we investigated the reactivity of β-O-4 models containing syringyl (S), guaiacyl (G), and *p*-hydroxy-phenyl (H) units (Fig. 3). For the depolymerization and bromination reactions, substrates with a G-type (left) and a methoxyl substituent (*ortho*-position, right) were successfully depolymerized and brominated (**1a** and **1c**), yielding dibrominated products **2a, 3a, 3c**, and **4a** in good to excellent yields. Substrates **1b** and **1d** also performed well and afforded **2a** in high yields. Notably, the right parts of **1b** and **1d** could be transformed into mono-brominated products **3b, 3d, and 4b**, which might be attributable to the low reactivity of bromo-substituted phenyl rings for further bromination. For the H-type models, **1e** and **1f** could smoothly afford mono-brominated products (**2b, 3e, and 4c**) and dibrominated product (**3c** and **4a**). The use of electron-rich S-type models **1g** and **1h** as substrates led to tribrominated product **2c** and corresponding mono-brominated products (**3b** and **4b**) and dibrominated product (**3a** and **4a**). Thus, for

this method, the degree of bromination could be controlled by the electronic properties of the lignin models. For depolymerization and chlorination reactions, various models exhibit high reactivity for depolymerization. The remaining part (G-type) of **1a-1d** could be completely transformed into tetra-chlorinated product **7a** in good to excellent yields (62-92%). However, owing to the strong electrophilic ability of Cl[+], mixed products from the right part of the **1a-1d** were found to have different degrees of chlorination (Supplementary Figs. 4-7). This method was also adapted for lignin model **1i** containing a hydroxyl group, affording corresponding halogenated products in moderate yields.

### Mechanistic investigations

To provide insights into the mechanism of this depolymerization and halogenation reaction, the activation effect of TFE was initially examined (Fig. 4a). The [1]H and [13]C NMR shifts of NBS with the addition of TFE were studied. The [1]H NMR signal of the methylene group in NBS shifted to a lower field in the presence of TFE, indicating that the electron density of the methylene group decreased. In addition, the [13]C NMR signal of the carbonyl group in NBS shifted to a lower field with the addition of TFE, revealing that the electron density of the carbonyl group decreased. A natural population analysis (NPA) for the electron density of NBS reveals that more positive charge distributes on the Br

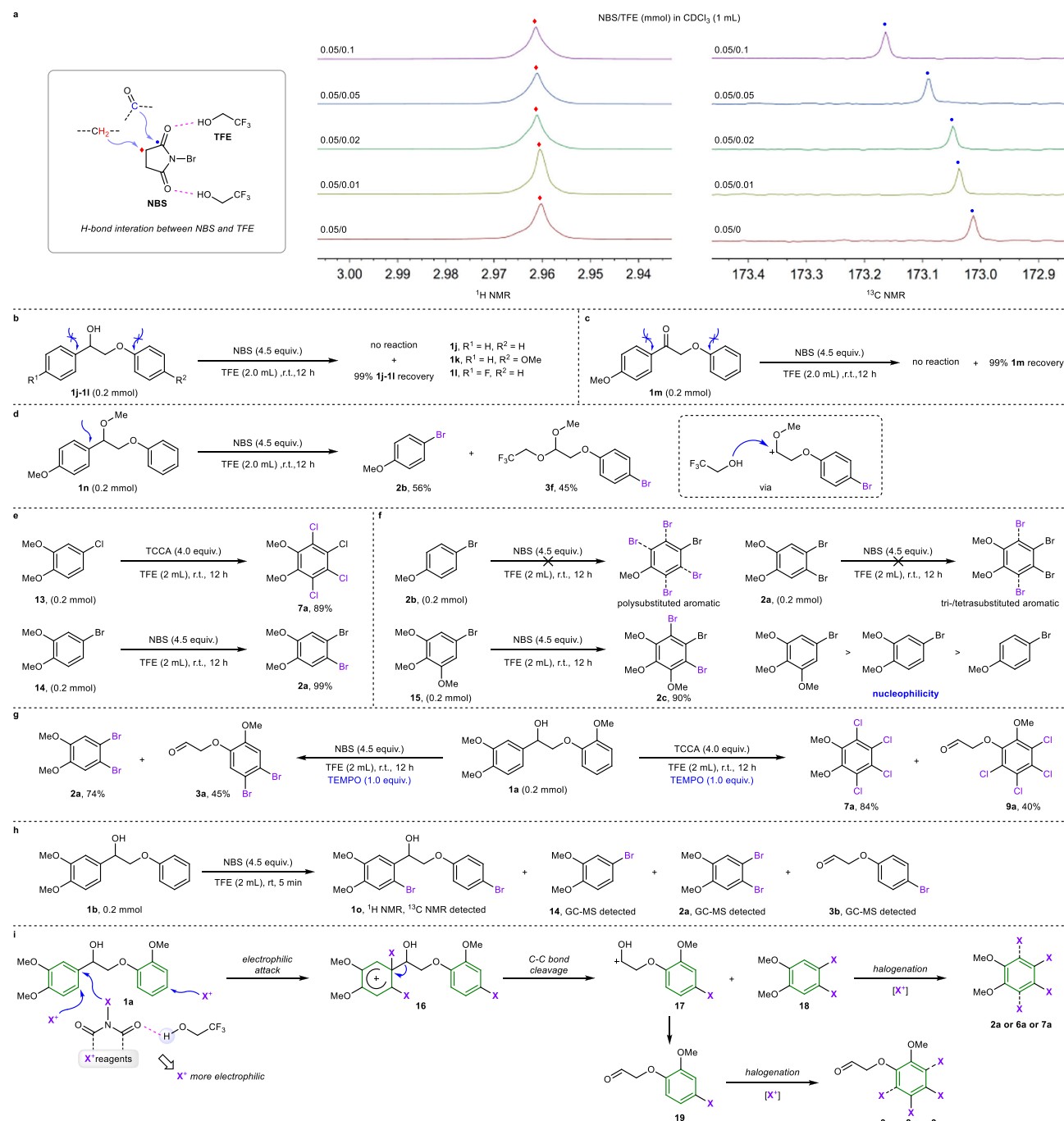

**Fig. 4 | Mechanism research. a** [1]H and [13]C NMR spectra study for halogenation reagent activation. **b** and **c** Reactivity of lignin linkages for research on bond cleavage. **d** Reaction of methyl protected lignin model for detection of intermediate. **e** Formation of polysubstituted aryl halides. **f** Interaction of electronic property of aromatics with reactivity of bromination. **g** Radical trapping experiments. **e** Proposed mechanism. **h** Bond cleavage and bromination research. **i** Proposed mechanism.

atom of TFE-activated NBS, which can explain the higher electrophilic reactivity (Supplementary Fig. 28). Next, the reactivity of lignin linkages with different electron properties was tested to investigate the behavior of bond cleavage by the electrophilic attack of halogen cations (Fig. 4b). Substrates with no substituent on the phenyl rings (**1j** and **1k**) or with a *para*-fluoro substituent (**1l**) exhibited inert reactivity, indicating that an electron-rich aromatic ring is crucial for the reaction. The reaction was also ineffective in oxidizing lignin models **1m** containing a $C_\alpha$=O group (Fig. 4c). Methyl-protected lignin model **1n** was used under standard reaction conditions, and depolymerization yielded brominated product **2b** in a 56% yield, indicating that the

hydroxyl group at the α position of the substrate is not necessary in this reaction (Fig. 4d). Notably, acetal product **3f** was obtained in a 45% yield, which was formed via the capture of carbocation intermediate by TFE. This result could prove the cleavage of C(sp²)−C(sp³) bond and the formation of carbocation intermediate. Moreover, the formation of polysubstituted aryl halides was confirmed by the reaction of mono-substituted 3,4-dimethoxy-1-chlorobenzene (**13**) and 3,4-dimethoxy-1-bromobenzene (**14**) under standard conditions (Fig. 4e). The reactivity of polybromination reactions was explored using mono-/di-/tri-methoxyl-substituted bromobenzene (**2a, 2b,** and **15**) as substrates under standard bromination reaction conditions (Fig. 4f). The result

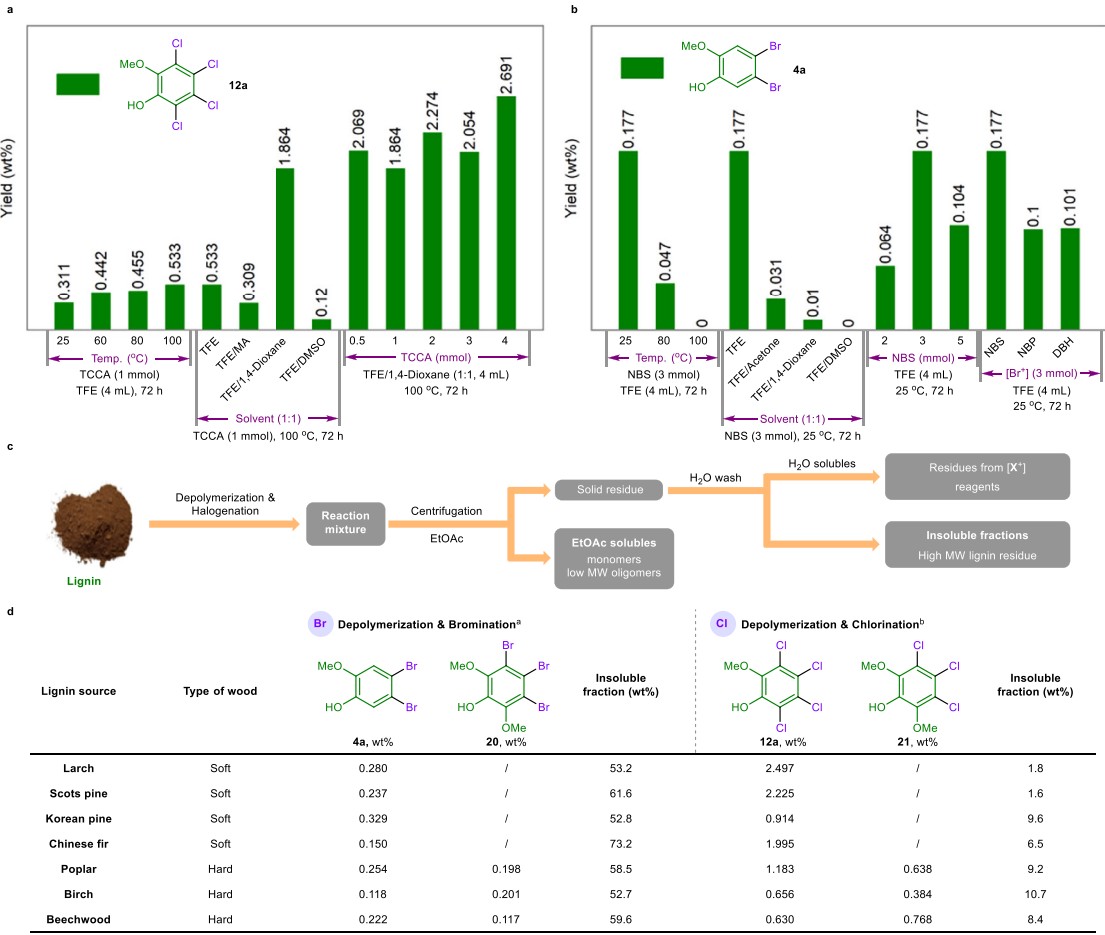

**Fig. 5 | Native lignin experiments. a** Screening conditions for depolymerization and chlorination. **b** Screening conditions for depolymerization and bromination. Larch lignin (50 mg) was used. **c** Standard process for separation and purification. **d** Depolymerization and halogenation of various lignins. [a]Reaction conditions: lignin (100 mg), NBS (6 mmol), TFE (10 mL), room temperature (25 °C), and 72 h. [b]Reaction conditions: lignin (100 mg), TCCA (8 mmol), TFE/1,4-dioxane (1:1, 10 mL), 100 °C, and 72 h. Products were analysed by GC-MS.

shows that **2a** and **2b** exhibited inert reactivities for further bromination, while trimethoxyl-substituted substrate **15** successfully produced tribrominated product **2c**. Notably, the result of bromination reactions for substrates with S-, G-, and H-type shows that the directing effect of methoxy group play an essential role for selective bromination, yielding brominated products with excellent para-selectivity (Fig. 3). Hence, the electronic property of substrates with S-, G-, and H-type directly affects the degree of overbromination and the directing effect of methoxyl group controls the site selectivity of bromination. Halogen cation reagents (N–X, where X = Cl, Br, or I) can produce nitrogen and halogen radicals[45], possibly triggering a radical-involving pathway. In this work, the addition of the free-radical scavenger 2,2,6,6-tetramethylpiperidinooxy (TEMPO) did not quench the depolymerization and halogenation reaction, indicating that a radical pathway can be excluded (Fig. 4g). To explore more details, reaction of **1b** under standard bromination conditions was detected within 5 min. Dibromination of substrate **1b** could be observed via the detection of **1o** and monosubstituted product **14** was also formed via the C(sp$^2$)–C(sp$^3$) bond cleavage of **1b** (Fig. 4h). This result indicates that the polyhalogenation and the carbon-carbon bond cleavage occurred simultaneously. Piecing together the above details and the reported relative mechanism[46], a possible reaction mechanism is proposed in Fig. 4i. According to this mechanism, the halogenation reagent is initially activated by TFE via hydrogen bond to generate a halogen cation, which subsequently performs a selective electrophilic attack on lignin model **1a** to afford intermediate **16**. Notably, a simultaneous

electrophilic substitution of halogen cation to benzene rings also proceeds. The C–C bond cleavage of intermediate **16** yields polyhalogenated product **18** and cation intermediate **17**. Further halogenation of compound **18** produces the final polyhalogenated product **2a**, **6a** or **7a**. The cation intermediate **17** is transformed into aldehyde **19**, which subsequently undergoes halogenation to yield polyhalogenated product **3a**, **8a** or **9a**.

## Depolymerization and halogenation of native lignin

After the lignin models were tested, we explored the application of this depolymerization and halogenation strategy in native lignin. Dioxasolv larch lignin was treated with TCCA as the cut-off and chlorination reagent in TFE at room temperature (25 °C), and a 0.311 wt% yield of tetrachlorinated product **12a** was successfully obtained (Fig. 5a). Increasing the reaction temperature to enhance the depolymerization process was further studied and the yield of monomer **12a** increased to 0.533 wt% at 100 °C. Considering the solubility of native lignin, mixed solvents were investigated to further increase the reaction efficiency. When the reaction was conducted in a mixed solvent of TFE and 1,4-dioxane (1:1, v/v), a 1.864 wt% yield of **12a** was obtained. Methyl acetate (MA) and dimethyl sulfoxide (DMSO) were also effective in this process when used as part of the mixed solvent, and decreased yields were obtained (0.309 and 0.12 wt%). Notably, increasing the dosage of TCCA significantly increased the yield of the target tetrachlorinated product **12a**, resulting in a 2.691 wt% yield when 4 mmol of TCCA was used in the reaction. The depolymerization and bromination of

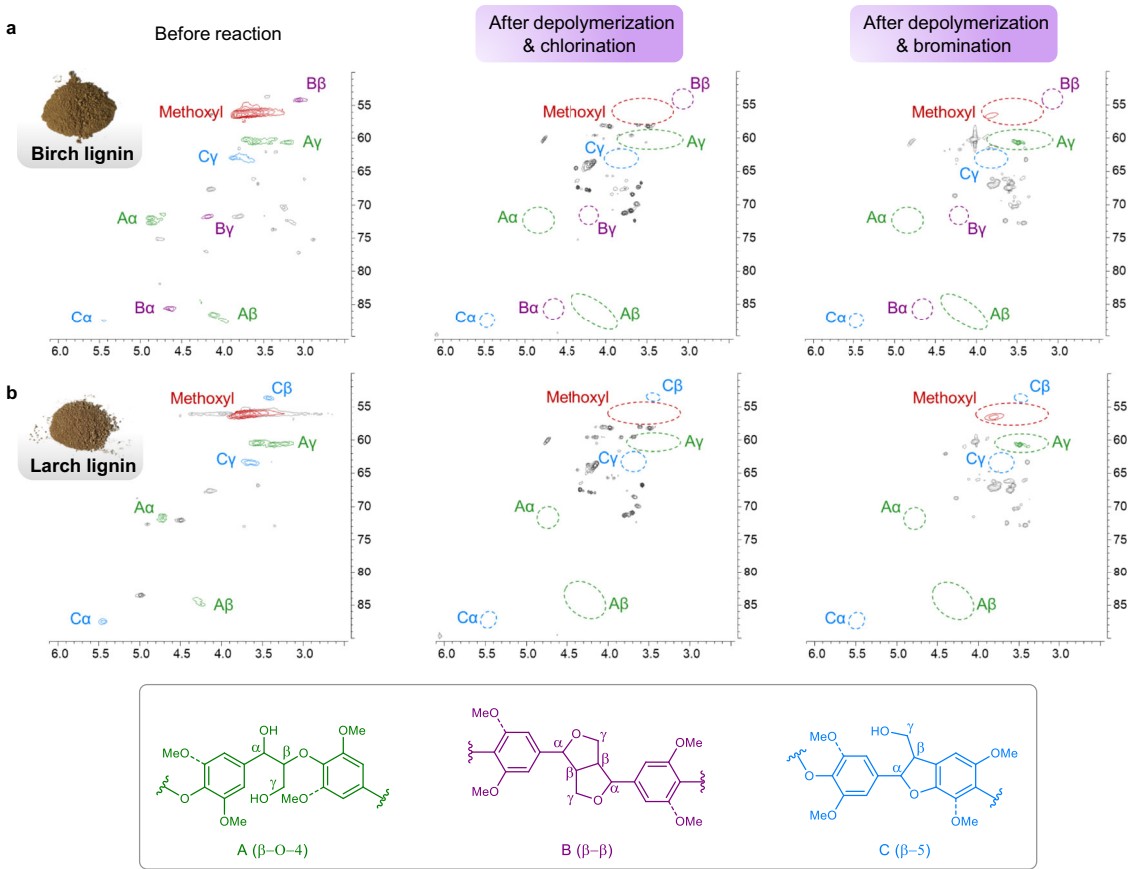

**Fig. 6 | HSQC spectra of lignin before and after the reaction. a** Birch lignin before and after depolymerization and halogenation. **b** Larch lignin before and after depolymerization and halogenation.

dioxasolv larch lignin were investigated under these conditions. As shown in Fig. 5b, increasing the reaction temperature had a negative effect on the reaction of lignin with NBS, and the highest yield of 0.177 wt% for dibrominated product **4a** was obtained. Unfortunately, further screening of the solvent, dosage of NBS, or species of Br⁺ reagent failed to increase the yield of the depolymerization and bromination process.

Under the optimized conditions and directed by the standard process for separation and purification (Fig. 5c), various lignins from softwoods (larch, Scots pine, Korean pine, and Chinese fir) and hard woods (poplar, birch, and beechwood) were used to test the developed method for the production of aryl halides. From the perspective of structural analysis, G-type units are the main components of lignin from soft woods, whereas in hard woods, both S- and G-type units exist. First, the reaction of lignin with NBS was investigated (Fig. 5d). Lignin from soft woods such as larch, Scots pine, and Korean pine exhibited good depolymerization performance and 38.4-47.2 wt% lignin was degraded, yielding 0.237-0.329 wt% brominated product **4a**. Chinese fir lignin showed lower reactivity for this transformation, affording 0.15 wt% target product with 73.2 wt% insoluble fraction. Compared with softwood-derived lignin, for lignin from hardwoods such as poplar, birch, and beechwood, a similar degree of degradation was achieved (40.4-47.3 wt%). The total yields of di- and tribrominated products **4a** and **20** were 0.452, 0.319, and 0.339 wt%, respectively. For depolymerization and chlorination reactions, both softwood- and hardwood-derived lignin exhibited higher degradation efficiencies than depolymerization and bromination of lignin did, and only 1.6-10.7 wt% of the insoluble fraction was recovered; this might be attributed to the higher reactivity of Cl⁺, which has a stronger interaction with lignin than Br⁺ does. Lignin from softwoods produced

tetrachlorinated product **12a** in 0.914-2.497 wt% yields. Moreover, lignin derived from hardwoods of poplar, birch, and beechwood generated tetra- and trichlorinated products **12a** and **21** in total yields of 1.821, 1.04, and 1.398 wt%, respectively. In addition, the gram scale experiments for depolymerization and halogenation of larch lignin were performed and showed higher efficiency than that of 100 mg scale. The result shows that 6.2 mg/g (0.62 wt%) and 35.7 mg/g (3.57 wt %) of dibrominated product **4a** and tetrachlorinated product **12a** were obtained using 1 g larch lignin as substrate, respectively, exhibiting potential practicability for producing aryl halides from lignin. Notably, the solvent and the residues (succinimide and cyanuric acid) from NBS and TCCA could be recollected by rotary evaporation and extraction, respectively (Supplementary Figs. 29 and 30).

The depolymerization and halogenation processes were also investigated via heteronuclear single quantum coherence spectroscopy (HSQC) NMR and gel permeation chromatography (GPC) experiments. The β-O-4, β-β, and β-5 linkages could be clearly identified by HSQC spectrum of dioxasolv birch lignin (Fig. 6a). The HSQC analysis of the soluble fraction after depolymerization and halogenation suggested a significant degree of depolymerization, as almost all the signals of Cα−H, Cβ−H, and Cγ−H of these three linkages were diminished (Fig. 6a). In dioxasolv larch lignin, β-O-4 and β-5 linkages could be observed via HSQC analysis (Fig. 6b). The disappearance of the Aα, Aβ, Aγ, Cα, Cβ, and Cγ signals revealed by HSQC spectra also confirmed a significant depolymerization process using our method (Fig. 6b). Moreover, compared to the original lignin from Birch and larch, GPC analysis of the soluble fraction after the reaction indicated a notable decrease in the molecular weight (MW) of the depolymerized lignin, and lower-MW oligomers and polymers could also be detected (Supplementary Figs. 19-26).

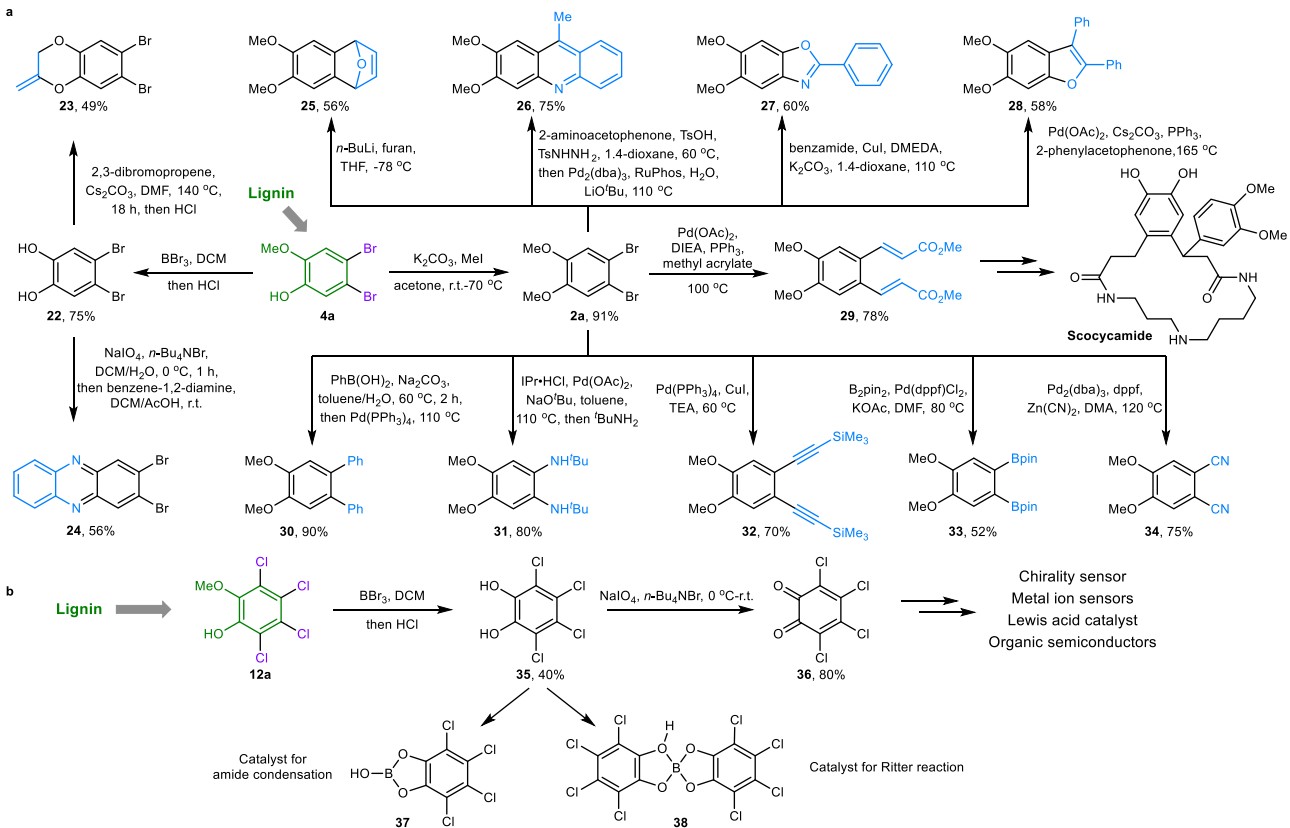

**Fig. 7 | Synthetic applications of lignin-derived aryl halides. a** Synthetic transformation of dibrominated product **4a. b** Synthetic transformation of tetrachlorinated product **12a**. Compounds **4a** and **12a** were obtained from commercial sources. DCM: dichloromethane. DMF: *N,N*-dimethylformamide. THF: tetrahydrofuran. Ts: *p*-toluenesulfonyl. DMEDA: *N,N'*-dimethyl-1,2-ethanediamine. DIEA: *N,N*-diisopropylethylamine. TEA: triethylamine. DMA: *N,N*-dimethylaniline. IPr·HCl: 1,3-bis (2,6-diisopropylphenyl) imidazolium chloride. dppf: 1,1'-bis(diphenylphosphino)ferrocene. B₂(pin)₂: bis(pinacolato)diboron.

## Synthetic application of lignin-derived aryl halides

As mentioned in the introduction, aryl halides are useful precursors and building blocks for chemical production, and further diverse synthetic transformations of lignin-derived dibromo- (Fig. 7a) or tetrachloro-substituted (Fig. 7b) aromatics were investigated to demonstrate their utility. First, the demethylation of dibrominated product **4a** from lignin could form 4,5-dibromobenzene-1,2-diol **22**, which could further afford heterocycle dioxine **23** and polymer precursor phenazine **24** via the transformation of the phenolic hydroxyl group. Then, cascade cyclization reactions of methylated product **2a** could generate heterocyclic motifs **25-28**. A two-step Heck reaction of **2a** with methyl acrylate could afford 1,2-phenylene product **29**, which is the starting material for the synthesis of the bioactive natural products scocycamide[47]. Further diarylation and diamination of **2a** could also be realized through Suzuki (**30**) and Buchwald-Hartwig (**31**) cross-coupling reactions. Notably, dialkynylated, diborated, and dicyanated products **32-34**, which are widely used as precursors for the synthesis of organic semiconductors[48], fused aromatics[49], and conjugated metal-organic frameworks (MOFs)[50], could be synthesized from dibromo product **2a**. Moreover, tetrachlorinated product **12a** could be converted into 3,4,5,6-tetrachlorobenzene-1,2-diol **35**, which can be further used for catalyst synthesis[51]. The simple oxidation of **35** could produce tetrachloro-*o*-benzoquinone **36**, which has been widely applied for the preparation of metal ion sensors[52], organic semiconductors[53], Lewis acid catalysts[54], and chirality sensors[55]. These results demonstrate the robust utility and synthetic applications of aryl halides from lignin.

## Discussion

In summary, we have developed an efficient and mild method for the depolymerization and halogenation of lignin into useful aryl halide compounds. Various wood-derived lignins can be converted into polybrominated and polychlorinated aromatics, which are useful precursors for diversiform synthetic applications in the preparation of functional molecules. The hydrogen bond activation of halogenation reagents by TFE is crucial for enhancing the reactivity of halonium ions (Br⁺ and Cl⁺). The electrophilic attack of halonium ions on the aromatic ring of lignin initiates the breakage of lignin linkages and further facilitates halogenation. This work represents a significant advancement in the development of simple and efficient protocols for lignin depolymerization and the production of value-added aromatics from lignin.

## Methods

### General procedure for depolymerization and bromination of lignin

A 25 mL sealed tube was charged with extracted native lignin (100 mg), NBS (6.0 mmol, 1.0678 g), and TFE (10 mL) under air atmosphere. The reaction was stirred for 72 h at room temperature. After the reaction is complete, the mixture was centrifuged with ethyl acetate (20 mL × 3) to separate the solid and liquid phases. The solid-phase fraction was stirred in 100 mL of water for 3 h to remove residues from the bromination reagent, and the insoluble fraction was collected by Brinell funnel filtration before drying and weighing. The liquid phase was concentrated in a vacuum and diluted with 30 mL of ethyl acetate, then extracted with water (50 mL × 3) to remove bromination reagent

residues from the solution. The organic phase was collected and dried with anhydrous sodium sulfate, then concentrated in a vacuum to quantify the product by GC-MS.

## General procedure for depolymerization and chlorination of lignin

A 25 mL sealed tube was charged with extracted native lignin (100 mg), TCCA (8.0 mmol, 1.8592 g) and TFE/1,4-dioxane (1/1, 10 mL) under air atmosphere. The reaction was stirred for 72 h at 100 °C. After the reaction is complete, the mixture is centrifuged with ethyl acetate (20 mL × 3) to separate the solid and liquid phases. The solid-phase fraction was stirred in 100 mL of water for 3 h to remove residues from the chlorination reagent, and the insoluble fraction was collected by Brinell funnel filtration before drying and weighing. The liquid phase was concentrated in a vacuum and diluted with 30 mL of ethyl acetate, then extracted with water (50 mL × 3) to remove chlorination reagent residues from the solution. The organic phase was collected and dried with anhydrous sodium sulfate, then concentrated in a vacuum to quantify the product by GC-MS.

## Data availability

Data generated in this study are provided in the main text and Supplementary Information files. The general information, optimization of reaction conditions, experimental procedures, and characterization of all compounds are provided in the Supplementary Information. All other data supporting the findings of this study are available within the paper and its Supplementary Information, or from the corresponding author upon request. Source data are provided with this paper.

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

## Acknowledgements

This work was supported by the Natural Science Funding of Heilong Jiang province for Excellent Young Scholar (YQ2022C004, B.T.), the National Natural Science Foundation of China (22378056, B.T.) and the Fundamental Research Funds for the Central Universities (2572022BB01, B.T., 2572023CT06, S.Liu).

## Author contributions

B.T., S.Liu., and Z.C. conceived the work and designed the experiments. Y.Liu. designed and performed the laboratory experiments. Y.Liu., Y.Li. Z.H., and S.W. collected the data. Y.Liu., Y.Li. Z.H. S.W. C.M. W.L., and S.Li. analyzed the data and wrote the manuscript. All authors discussed the results and commented on the manuscript.

## Competing interests

The authors declare no competing interests.
