## [Transparent Peer Review file · Nature Communications]

Producing aryl halides from lignin

Corresponding Author: Professor Bing Tian

Version 0:

Reviewer comments:

Reviewer #1

(Remarks to the Author)

In this manuscript, the authors reported a new strategy to depolymerize and produce the aryl halides through TFE promoted electrophilic substitution of halogen. The method has the advantages of high selectivity, mild condition and usefulness, affording various aryl halides as the precursors for the preparation of chemicals, pharmaceuticals and catalysts. More importantly, the reaction also works well with native lignin, and a thorough mechanistic study has been provided. The authors have applied the dehydroxymethyl bromination reaction to the lignin depolymerization, expanding the scope of high value chemicals that can be produced from biomass, and providing a new strategy for the utilization of lignin. Therefore, I recommend acceptance of this work by Nature Communications after below mentioned comments are considered by authors.

(1) The depolymerization of lignin is based on the TFE promoted dehydroxymethyl halogenation reaction, which was reported by Shibata et. al. with a similar condition and the same mechanism (Synlett 2018, 29, 2275-2278, DOI: 10.1055/s-0037-1610980). Therefore, this work should be cited and discussed.

(2) In Fig. 2b, entry 6, an excellent para-selectivity was obtained for the bromination reaction (with a 99% yield of 2a obtained), which is very important to the cleavage of the C(sp²)-C(sp³) bond. The authors should highlight this point and explain the reason of the excellent regioselectivity. For other entries like 3,4,5,7,8, when the yields of 2a are lower, is it because the conversion is lower, or tri- and tetra-brominated products are formed, or other byproducts are formed? If tri- and tetra-brominated products are formed, the yields should be provided as Fig. 2c.

(3) In Fig. 2b, entry 6, 99% of 2a and 63% of 3a were obtained, why the yield of 3a is much lower than 2a (Fig. 2c, entries 7 and 8 also have the same situation)? The byproducts of 3a and 7a' should be identified.

(4) In fig. 5, the reactions are applied to native lignins, but the yields are very low, especially for the bromination reaction. The authors should explain the reason why it is so different from the model compounds. Normally, the hydroxyl in the native lignin will have a strong influence on the depolymerization reactions, but the model compounds only with methoxyl are provided. Therefore, I suggest the authors to add some model compounds containing hydroxyl to study the effect of hydroxyl.

(5) For the depolymerization of native lignins, the products were analyzed by GC-MS (Fig. 5b). How to separate and purify this product from the mixture with a large numbers of byproducts derived from lignins? The detailed procedure should be provided.

(6) In Fig. 7, further diverse synthetic transformations of lignin-derived aromatics 14 and 16 were investigated. Are 14 and 16 obtained from the depolymerization of native lignin and the subsequent purification, or obtained from the commercial source? The authors should clarify this in the legend of Fig. 7.

(7) In Supplementary Table 1 and Table 2, why 2a was determined by ¹H NMR while 3a was determined by GC-MS, which is different from the manuscript (Fig. 2b)?

(8) Some mistakes should be corrected:

- In page 2, line 48, 'develop' should be changed to 'developing'
- In page 4, line 70, 'Lignin model 1a was treated with NBS' should be changed to 'Treating lignin model 1a with NBS'
- In page 11, line 192, 'while' should be changed to 'While'
- In page 14, line 234, 'Further di-arylation and di-arylation of 21' should be changed to 'Further di-arylation and di-amination of 21'
- In page 17, line 301, 'N-Heteroacenes and N-heteroarenes' should be changed to 'N-heteroacenes and N-heteroarenes'
- In Supplementary Table 8, '0280' should be changed to '0.280' for the yield of 14 with Larch lignin.

Reviewer #2

(Remarks to the Author)

This manuscript describes that the production of aryl halides were efficiently produced through depolymerization and halogenation of lignin with N-halosuccinimide and 2,2,2-trifluoroethanol. It is found that the hydrogen bonding activation for halogenation reagents is essential for substantially increasing the reactivity, resulting a highly efficient cleavage of C–C bonds in lignin. The method realized the precise depolymerization and halogenation reactions from lignin models to native lignin from various wood resources and provided a promising approach for the synthesis of useful aryl halides. It can be acceptable after some major revisions.

Several questions and suggestion are given in the following.

1) For depolymerization and bromination process, the main obtained product is di-brominated aromatic molecule; however, for depolymerization and chlorination process, the tetra-chlorinated aromatics are majorly generated. Why is this phenomenon?

2) How can the C=C double bond be formed in the compound 3d which generates from the depolymerization and bromination? Moreover, when methyl-protected lignin model was reacted with N-bromosuccinimide, what is the other product except for the brominated product 2b during reaction?

3) It is suggested that the author should perform the scale-up experiments, at least > gram level experiment in order to present more useful data for the industrial application.

4) The references on the valorization of lignin (such as The Journal of Organic Chemistry, 2013, 78, 18, 9431-9443; ACS Sustainable Chemistry & Engineering, 2023, 11, 28, 10203-10218) should be supplemented in text.

In addition, the English expressions including the grammar need to be further improved to make it readable and understandable.

Reviewer #3

(Remarks to the Author)

In this manuscript, the authors reported a depolymerization and halogenation approach of degrading lignin into aryl halide compounds by using NBS or its analogs as halogenating reagents and TFE as solvents. Although encouraging results have been obtained from the current research, the detailed mechanism was suggested to be given before its acceptance based on the following.

1. The reaction mechanism needs more convincing supports. For examples, more control experiments or calculations are suggested to prove the crucial importance of TFE as an effective solvent. Besides, mechanism shall hypothesize whether polyhalogenation occurs prior to the carbon-carbon bond cleavage or they act simultaneously.

2. It is suggested that the authors refine the synthetic methods to validate its synthetic value.

The current synthetic methods are not of sufficient synthetic value as 100 mg of lignin consume large quantity of the halogenating reagents.

3. The authors highlighted the importance of the obtained halogenated aromatics as useful precursors for synthesis of other useful organic compounds. However, compound 21 is a readily available organic compound and many of its derivatized reactions have been well established. Derivatization of the halogenated aromatics that are depolymerized and halogenated by the approach shall be paid attention rather than commercially available or readily accessed by a simple transformation.

Version 1:

Reviewer comments:

Reviewer #1

(Remarks to the Author)

In the revised manuscript, all the concerns have been addressed. Therefore, I recommend acceptance of this work without further revision.

Reviewer #3

(Remarks to the Author)

1. It is helpful to add the part about the effect of TFE for promotion the reactivity of NBS, some control experiments conducted, and the revised mechanism for this depolymerization and halogenation reaction.

2. The scale-up experiment is very challenging.

3. The mechanism is more clearly with the revised clarification in the legend.

Response to Reviewers Comments

Reviewer 1:

In this manuscript, the authors reported a new strategy to depolymerize and produce the aryl halides through TFE promoted electrophilic substitution of halogen. The method has the advantages of high selectivity, mild condition and usefulness, affording various aryl halides as the precursors for the preparation of chemicals, pharmaceuticals and catalysts. More importantly, the reaction also works well with native lignin, and a thorough mechanistic study has been provided. The authors have applied the dehydroxymethyl bromination reaction to the lignin depolymerization, expanding the scope of high value chemicals that can be produced from biomass, and providing a new strategy for the utilization of lignin. Therefore, I recommend acceptance of this work by Nature Communications after below mentioned comments are considered by authors.

1. The depolymerization of lignin is based on the TFE promoted dehydroxymethyl halogenation reaction, which was reported by Shibata et. al. with a similar condition and the same mechanism (Synlett 2018, 29, 2275-2278, DOI: 10.1055/s-0037-1610980). Therefore, this work should be cited and discussed.

Response: We thank you for your nice suggestion. This reference reported an in-situ production of Br^+ by reaction of strong oxidant $\text{PhI}(\text{OAc})_2$ with Br^- . We have cited this reference and added the discussion in the main text of revised manuscript. (Mechanistic investigations section, described as “Piecing together the above details and the reported relevant mechanism⁴⁷, a possible reaction mechanism is proposed in Fig. 4i.”)

2. In Fig. 2b, entry 6, an excellent para-selectivity was obtained for the bromination reaction (with a 99% yield of 2a obtained), which is very important to the cleavage of the $\text{C}(\text{sp}^2)$ - $\text{C}(\text{sp}^3)$ bond. The authors should highlight this point and explain the reason of the excellent regioselectivity. For other entries like 3,4,5,7,8, when the yields of 2a are lower, is it because the conversion is lower, or tri- and tetra-brominated products are formed, or other byproducts are formed? If tri- and tetra-brominated products are formed, the yields should be provided as Fig. 2c.

Response: We thank you for your constructive suggestion which are quite important for the reaction. Two main reasons could be proposed for the selectivity:

1) The aromatic ring at left part of the model **1a** is more electron-rich than that of the right part, thus highly selective electrophilic attack of Br^+ to aromatic ring occurs at $\text{C}(\text{sp}^2)$ - $\text{C}(\text{sp}^3)$ bond (left part). We have added the corresponding description in the revised manuscript as: “Increasing the dosage of TFE further promoted the reaction (Fig. 2b, entries 4 and 5). Notably, yields of 99% for **2a** and 61% for **3a** were obtained (Fig. 2b, entry 6) when TFE was used as the sole solvent, indicating efficient cleavage of the $\text{C}(\text{sp}^2)$ - $\text{C}(\text{sp}^3)$ bond with excellent selectivity.” The reaction reactivity was also explained by the control experiments in Fig. 4b-4f, which the active site of electrophilic attack of Br^+ is decided by the electrical property of aromatic rings.

Figs. 4b-4f

2) The cleavage of the $C(sp^2)-C(sp^3)$ bond is the main driving force for the reaction. The cleavage of $C(sp^2)-C(sp^3)$ bond rather than the cleavage of $C(sp^2)-O$ bond could lead to the high selectivity.

We thank you for your careful observation. For entries like 3,4,5,7,8 in Fig 2b, when the yields of 2a are lower, because the conversion is lower, we have added the conversions of all the entries in the revised Supplementary Information. In addition, we rechecked the GC-MS for all the experiments of condition screening and the substrate scope, and the phenol compounds **4a** and **10a-12a** were detected and isolated, which were formed by the decomposition of **3a** and **8a/9a**. We have added the yields of phenol products in all the conditions of reaction screening (Fig. 2b and 2c) and lignin models (Fig. 3) in the revised manuscript.

Figs. 2b and 2c

The tri- and tetra-brominated products are not formed for substrates with one or two -OMe substituted in the aromatic rings (G- and H- type linkage). This result was also verified by the reaction of **2a** and **2b** under standard bromination conditions, which no further bromination was occurred (Fig. 4f). While for the S-type linkage (with 3 -OMe) substrate **1g** and **1h**, tribrominated products could form (Fig. 3a) and also be verified by the reaction of 1-Bromo-3,4,5-trimethoxybenzene **15**. This could also be the evidence of the electronic property of substrates with S-, G-, and H-type could directly affect the degree of overbromination. We have added all the above results and discussions in the revised manuscript.

Fig. 4f

Fig. 3a

3. In Fig. 2b, entry 6, 99% of 2a and 63% of 3a were obtained, why the yield of 3a is much lower than 2a (Fig. 2c, entries 7 and 8 also have the same situation)? The byproducts of 3a and 7a' should be identified.

Response: We thank you for your nice suggestion. As we mentioned in question 2, we rechecked the GC-MS for all the experiments of condition screening and the substrate scope, and the phenol compounds **4a** and **10a-12a** were detected and isolated (character by ^1H and ^{13}C NMR and high-resolution mass spectrometry (HRMS), in Supplementary Information). We have added the yields of phenol products in all the conditions of reaction screening (Fig. 2b and 2c) and lignin models (Fig. 3) in the revised manuscript.

b Conditions optimization for depolymerization and bromination

Entry	Solvent	$[\text{Br}^+]$	Yield of 2a (%)	Yield of 3a/4a (%)
1	Acetone	NBS	0	0/0
2	1,4-Dioxane	NBS	0	0/0
3	Acetone/TFE (9:1)	NBS	51	32/10
4	Acetone/TFE (5:1)	NBS	63	36/11
5	Acetone/TFE (1:1)	NBS	73	47/14
6	TFE	NBS	99	61/15
7	TFE	NBP	83	40/16
8	TFE	DBDMH	77	33/14
9	TFE	NDBI	5	trace/7

c Conditions optimization for depolymerization and chlorination

Entry	Solvent	$[\text{Cl}^+]$	Yield of 5a (%)	Yield of 6a (%)	Yield of 7a (%)	Yield of 8a/9a (%)	Yield of 10a/11a/12a (%)
1	TFE	NCS	3	0	0	0/0	0/0/0
2	TFE	DCDMH	2	0	0	0/0	0/0/0
3	TFE	Na-DCC	11	0	0	0/0	0/0/0
4	TFE	TCCA	48	22	0	32/0	18/0/0
5	TFE/Methyl acetate (1:1)	TCCA	58	30	0	22/0	9/6/0
6	TFE/Methyl acetate (1:1)	TCCA (2.5 equiv.)	28	50	0	28/0	16/9/0
7	TFE	TCCA (2.5 equiv.)	0	0	78	10/14	3/5/10
8	TFE	TCCA (4.0 equiv.)	0	0	92	0/34	0/0/20

4. In fig. 5, the reactions are applied to native lignins, but the yields are very low, especially for the bromination reaction. The authors should explain the reason why it is so different from

the model compounds. Normally, the hydroxyl in the native lignin will have a strong influence on the depolymerization reactions, but the model compounds only with methoxyl are provided. Therefore, I suggest the authors to add some model compounds containing hydroxyl to study the effect of hydroxyl.

Response: We appreciate your valuable suggestion. We synthesized model compound **1i** with a hydroxyl substitution and tested in the bromination and chlorination reactions. As shown below, this method was also effective to hydroxyl containing substrate. For chlorination reactions (71% conversion), products **10a**, **11a**, and **12a** which different degrees of chlorination were isolated in a total 60% yield, and yields of 15% and 9% for aldehyde products **8a** and **9a**, respectively. For bromination reactions (53% conversion), the reaction was also effective albeit with a lower reactivity. 49% yield of **4a** and 18% of **3a** could be obtained. We have added this result in the revised manuscript (Fig. 3).

5. For the deloymerization of native lignins, the products were analyzed by GC-MS (Fig. 5b). How to separate and purify this product from the mixture with a large numbers of byproducts derived from lignins? The detailed procedure should be provided.

Response: Thank you for your suggestions. For 100 mg scale lignin reactions, preparative-thin-layer-chromatography (PTLC, 20×20 mm) was used for isolated the halogenated products. For gram scale experiment, the silica gel column chromatography was used for isolation of products. We added the procedure in the revised Supplementary Information (Figs. 29 and 30).

6. In Fig. 7, further diverse synthetic transformations of lignin-derived aromatics **14** and **16** were investigated. Are **14** and **16** obtained from the depolymerization of native lignin and the subsequent purification, or obtained from the commercial source? The authors should clarify this in the legend of Fig. 7.

Response: The synthetic transformation of halogenated compounds from lignin aimed to show the potential practicality of the poly-halogenated products from the view of synthetic chemistry, and established a route from lignin to diverse functional molecules. For more efficient exploration of the downstream applications of lignin-derived products, the compounds **14** (revised **4a**) and **16** (revised **12a**) in Fig.7 are obtained from commercial source, we added the clarification in the legend of Fig. 7 as “Compounds **4a** and **12a** were obtained from commercial sources”. Notably, the scale-up experiments show that 1 gram-scale reaction (35.7 mg **12a** from 1 g larch lignin, 6.2 mg **4a** from 1 g larch lignin, isolated yields) was also effective and exhibited higher reactivity than that of 100 mg scale reaction, showing potential production of **4a** and **12a** from real lignin.

Supplementary Figure 29. Gramm scale experiment for depolymerization and bromination of larch lignin.

Supplementary Figure 30. Gramm scale experiment for depolymerization and chlorination of larch lignin.

7. In Supplementary Table 1 and Table 2, why 2a was determined by ¹H NMR while 3a was determined by GC-MS, which is different from the manuscript (Fig. 2b)?

Response: We appreciate your comments. For condition screening, ¹H NMR was mainly used as an efficient tool for detection of reaction conversion and the yield of 2a and 5a-9a due to their distinct characteristic peaks. The yields of 3a, 4a, and 10a-12a were detected via GC-MS analysis of the unpurified reaction mixture with *n*-octadecane and *n*-dodecane as an internal standard, respectively. The method for determination of reaction in Supplementary Table 1-4 is same with Fig. 2b and 2c. We have corrected the information in revised manuscript and Supplementary Information.

8. Some mistakes should be corrected:

- In page 2, line 48, 'develop' should be changed to 'developing'
- In page 4, line 70, 'Lignin model 1a was treated with NBS' should be changed to 'Treating lignin model 1a with NBS'
- In page 11, line 192, 'while' should be changed to 'While'
- In page 14, line 234, 'Further di-arylation and di-arylation of 21' should be changed to 'Further di-arylation and di-amination of 21'
- In page 17, line 301, 'N-Heteroacenes and N-heteroarenes' should be changed to 'N-heteroacenes and N-heteroarenes'
- In Supplementary Table 8, '0280' should be changed to '0.280' for the yield of 14 with Larch lignin.

Response: We thank you for your careful view and nice suggestion. We have corrected all the mentioned mistakes and also double-checked the whole manuscript and Supplementary Information.

Reviewer 2:

This manuscript describes that the production of aryl halides were efficiently produced through depolymerization and halogenation of lignin with N-halosuccinimide and 2,2,2-trifluoroethanol. It is found that the hydrogen bonding activation for halogenation reagents is essential for substantially increasing the reactivity, resulting a highly efficient cleavage of C–C bonds in lignin. The method realized the precise depolymerization and halogenation reactions from lignin models to native lignin from various wood resources and provided a promising approach for the synthesis of useful aryl halides. It can be acceptable after some major revisions.

Several questions and suggestion are given in the following.

1. For depolymerization and bromination process, the main obtained product is di-brominated aromatic molecule; however, for depolymerization and chlorination process, the tetra-chlorinated aromatics are majorly generated. Why is this phenomenon?

Response: Thanks for the comment and advice. The halogenation degree of this reaction is mainly affected by two factors: 1) The reactivity of Br^+ and Cl^+ with aromatic molecule is mainly determined by their electrophilicity. 2) The electronic property of aromatic substrate. For this work, firstly, compared to Br^+ , the Cl^+ is more reactive as the Cl^+ (smaller atom, larger electronegativity) shows stronger electrophilicity than Br^+ , leading to more reactive chlorination. As shown in Fig. 3 in main text, for 1 -OMe substituted substrates **1e** and **1f**, only monosubstituted product **2b** could be obtained. For 2 -OMe substituted substrates **1a-1d**, dibromination products were formed and for 3 -OMe substituted substrates **1g** and **1h** tribromination products were yielded. This result shows that the degree of bromination corresponds to the electronic property of substrate with different substitutions. Moreover, control experiments in Fig. 4e and 4f show that 4-methoxy-1-bromobenzene **2b** could not be further brominated. 3,4-Dimethoxy-1-bromobenzene **14** could undergo a bromination to yield dibrominated product **2a**, which further bromination can not occur. 3,4,5-Trimethoxy-1-bromobenzene **15** can afford tribrominated product **2c**. These control experiments show the same result with experiments using mono-, di-, or tri-methoxyl substituted substrates in Fig. 3.

Hence, the electronic property of substrates with S-, G-, and H-type directly affects the degree of overbromination.

For chlorination reaction of 3,4-dimethoxy-1-chlorobenzene (**13**), poly-chlorination occurred due to the strong electrophilicity of Cl⁺.

Fig. 4e and 4f

2. How can the C=C double bond be formed in the compound **3d** which generates from the depolymerization and bromination? Moreover, when methyl-protected lignin model was reacted with *N*-bromosuccinimide, what is the other product except for the brominated product **2b** during reaction?

Response: The C=C double bond was formed through the dehydration of the alcohol intermediate. This C=C double bond-containing product could usually be obtained in lignin depolymerization through the dehydration of similar alcohol intermediate (see refs.: (1) *ACS Catal.* 2024, 14, 11733–11740, DOI: 10.1021/acscatal.4c03469; (2) *Angew. Chem. Int. Ed.* 2012, 51, 3410–3413, DOI: 10.1002/anie.201107020; (3) *ACS Catal.* 2016, 6, 4399–4410, DOI: 10.1021/jo401680z; (4) *Inorg. Chem.* 2012, 51, 7354–7361, DOI:10.1021/ic3007525).

The reaction of methyl-protected lignin model (**1n**) under standard conditions, 56% yield of brominated product **2b** and 45% yield of product **3f** could be obtained (isolated and characterized by ¹H and ¹³C NMR and HRMS) which was formed via the capture of carbocation intermediate by TFE. This could also be important evidence for detection the key intermediate of the reaction and could be conducive to further understanding the mechanism. We have added this result in the revised manuscript (Fig. 5d).

3. It is suggested that the author should perform the scale-up experiments, at least > gram level experiment in order to present more useful data for the industrial application.

Response: We thank you for your nice suggestion. We conducted the gram scale experiments for depolymerization and bromination or chlorination of real lignin. Comfortingly, the result showed that 6.2 mg (0.62 wt%, isolated by silica gel column chromatography) of dibrominated

product **4a** and 35.7 mg (3.57 wt%, isolated by silica gel column chromatography) of tetrachlorinated product **12a** could be obtained from the reactions of 1 g larch lignin, exhibiting higher reactivity than that of 100 mg scale. Notably, the residue of halogenating reagents (succinimide from NBS, cyanuric acid from TCCA) could be recollected in 90 mol% and 87 mol % yields, respectively, which are used as the starting materials for production of NBS and TCCA. This could be a support for solving the problems of halogenating reagents consume during this lignin transformation. The procedure for gram scale reaction was added in the revised Supplementary Information (Supplementary Figs. 29 and 30) as follows:

Supplementary Figure 29. Gram scale experiment for depolymerization and bromination of larch lignin.

Supplementary Figure 30. Gram scale experiment for depolymerization and chlorination of larch lignin.

4. The references on the valorization of lignin (such as *The Journal of Organic Chemistry*, 2013, 78, 18, 9431-9443; *ACS Sustainable Chemistry & Engineering*, 2023, 11, 28, 10203-10218) should be supplemented in text.

Response: We thank you for your comments. We have added these references in the revised manuscript.

5. In addition, the English expressions including the grammar need to be further improved to make it readable and understandable.

Response: We thank you for your nice suggestion. We used language editing service of Springer Nature to improve the language level of the main text, we hope the improvement could make the revised manuscript clearer for readers.

Reviewer 3:

In this manuscript, the authors reported a depolymerization and halogenation approach of degrading lignin into aryl halide compounds by using NBS or its analogs as halogenating reagents and TFE as solvents. Although encouraging results have been obtained from the current research, the detailed mechanism was suggested to be given before its acceptance based on the following.

1. The reaction mechanism needs more convincing supports. For examples, more control experiments or calculations are suggested to prove the crucial importance of TFE as an effective solvent. Besides, mechanism shall hypothesise whether polyhalogenation occurs prior to the carbon-carbon bond cleavage or they act simultaneously.

Response: We thank you for your insightful suggestion. To better understanding the effect of TFE for promotion the reactivity of NBS, a natural population analysis for the electron density of NBS with/without TFE was calculated, the result shows that more positive charge distributes on the Br atom of TFE-activated NBS, which can explain the higher electrophilic reactivity. We have added this result in the revised manuscript as “*A natural population analysis (NPA) for the electron density of NBS reveals that more positive charge distributes on the Br atom of TFE-activated NBS, which can explain the higher electrophilic reactivity (Supplementary Fig. 28)*”. The calculation result was added in the revised Supplementary Information (Supplementary Fig. 28) and shown as follows:

In addition, some control experiments were conducted for further understanding the halogenation reactions. First, the key intermediate was observed using methylated substrate **1n**, the product **3f** was isolated in 45% yield, which was formed via the capture of carbocation intermediate by TFE. This result could prove the cleavage of C(sp²) – C(sp³) bond and the formation of carbocation intermediate. We have added this result in the revised manuscript (Fig. 5d).

Fig. 5d

The reactivity of polybromination reactions was explored using mono-/di-/trimethoxyl-substituted bromobenzene (**2a**, **2b**, and **15**) as substrates under standard bromination reaction conditions (Fig. 4f). The result shows that **2a** and **2b** exhibited inert reactivities for further bromination, while trimethoxyl-substituted substrate **15** successfully produced tribrominated product **2c**. Hence, the electronic property of substrates with S-, G-, and H-type directly affects the degree of overbromination. We have added this result in the revised manuscript (Figs. 5e and 5f).

Figs. 5e and 5f

Moreover, as the reviewer mentioned, owing to the electron donating effect of -OMe in the lignin linkage, the electrophilic aromatic halogenation and the carbon-carbon bond cleavage might be occurred simultaneously. To explore more details, reaction of **1b** under standard bromination conditions was detected within 5 min. Dibromination of substrate **1b** could be observed via detection of **1b'** and monosubstituted product **14** was also formed via the C(sp²)–C(sp³) bond cleavage of **1b**. This result indicates that the polyhalogenation and the carbon-carbon bond cleavage occurred simultaneously. We have added this result in the revised manuscript (Fig. 4h).

Fig.4h

Therefore, we revised the description of possible mechanism for this depolymerization and halogenation reaction in the revised manuscript as follows: “Piecing together the above details and the reported relative mechanism, a possible reaction mechanism is proposed in Fig. 4i. According to this mechanism, the halogenation reagent is initially activated by TFE via hydrogen bond to generate a halogen cation, which subsequently performs a selective

electrophilic attack on lignin model **1a** to afford intermediate **16**. Notably, a simultaneous electrophilic substitution of halogen cation to benzene rings also proceeds. The C–C bond cleavage of intermediate **16** yields polyhalogenated product **18** and cation intermediate **17**. Further halogenation of compound **18** produces the final polyhalogenated product **2a** or **6a**. The cation intermediate **17** is transformed into aldehyde **19**, which subsequently undergoes halogenation to yield polyhalogenated product **3a**, **8a** or **9a**.”

Fig.4i

We believe that these revisions have significantly strengthened the possible mechanism. We hope that these improvements will make the mechanism clearer.

2. It is suggested that the authors refine the synthetic methods to validate its synthetic value. The current synthetic methods are not of sufficient synthetic value as 100 mg of lignin consume large quantity of the halogenating reagents.

Response: We thank you for the suggestions. The scale-up experiment for chemical production especially those of high valued molecules is still very challenging in the area and is of great significant. Thus, we also tested gram-scale experiments for halogenation and depolymerization of lignin. The result showed that 6.2 mg (0.62 wt%, isolated by silica gel column chromatography) of dibrominated product **4a** and 35.7 mg (3.57 wt%, isolated by silica gel column chromatography) of tetrachlorinated product **12a** could be obtained from 1 g larch lignin. The reactivity of scale-up experiments was higher than that of 100 mg scale. Notably, the residue of halogenating reagents (succinimide from NBS, cyanuric acid from TCCA) could be recollected in 90% and 87% yields, respectively, which are used as the starting materials for production of NBS and TCCA. This could be a support for solving the problems of halogenating reagents consume during this lignin transformation. We know that there is a long way to go for practical application of production of fine chemicals from real lignin, we here show that the new route of lignin refinery for production of value-added aryl halides which vary from the conventional products.

We have added the following sentence in the revised manuscript: “*In addition, the gram scale experiments for depolymerization and halogenation of larch lignin were performed and showed higher efficiency than that of 100 mg scale. The result shows that 6.2 mg/g (0.62 wt%) and 35.7 mg/g (3.57 wt%) of dibrominated product **4a** and tetrachlorinated product **12a** were obtained using 1 g larch lignin as substrate, respectively, exhibiting potential practicability for producing aryl halides from lignin. Notably, the solvent and the residues of NBS (succinimide) and TCCA (cyanuric acid) could be recollected by rotary evaporation and extraction, respectively (Supplementary Figs. 29 and 30).*”

The procedure for gram scale reaction was added in the revised Supplementary Information as follows:

Supplementary Figure 29. Gramm scale experiment for depolymerization and bromination of larch lignin.

Supplementary Figure 30. Gramm scale experiment for depolymerization and chlorination of larch lignin.

3. The authors highlighted the importance of the obtained halogenated aromatics as useful precursors for synthesis of other useful organic compounds. However, compound 21 is a readily available organic compound and many of its derivatized reactions have been well established. Derivatization of the halogenated aromatics that are depolymerized and halogenated by the approach shall be paid attention rather than commercially available or readily accessed by a simple transformation.

Response: We appreciate your valuable suggestions. We highly agree with the referee's point "derivatization of the halogenated aromatics that are depolymerized and halogenated by the approach shall be paid attention rather than commercially available or readily accessed by a simple transformation". The synthetic transformation of halogenated compounds from lignin aimed to discover the potential practicality of the poly-halogenated products in synthetic chemistry, and show an entire line from lignin to diverse functional molecules. For more efficient exploration of the downstream applications of lignin-derived products, the compounds

14 (revised **4a**) and 16 (revised **12a**) in Fig.7 are obtained from commercial source, we added the clarification in the legend of Fig. 7 as “*Compounds 2a and 14a were obtained from commercial sources*”. Notably, as mentioned in question 2, the scale-up experiments show that gram-scale reaction (35.7 mg **4a** from 1 g Larch lignin, 6.2 mg **12a** from 1 g Larch lignin, isolated yields, in revised Supplementary Information Figs. 29 and 30) was also effective and exhibited higher reactivity than that of 100 mg scale reaction, showing potential production of **4a** and **12a** from real lignin.

Point-by-point Response to Reviewers' Comments

Reviewer #1 (Remarks to the Author):

In the revised manuscript, all the concerns have been addressed. Therefore, I recommend acceptance of this work without further revision.

Response: We thank Reviewer 1 for positive comments on our revision work and support for the publication of our manuscript.

Reviewer #2 (Remarks to the Author):

[Reviewer 2 was unable to continue the review process with us. As such, we asked Reviewer 1 to assess if their concerns had been addressed in revision. Please see the attached file.]

For question (1), I agree that poly-chlorination occurred due to the stronger electrophilicity of Cl^+ than Br^+ . However, I believe the main reason for the different bromination degree is the excellent para-selectivity of bromination reaction within this particular reaction system. This selectivity ensures that the bromination occurs exclusively at the para-position relative to the methoxyl group. As shown in Fig. 3 in main text, for 1 -OMe substituted substrates 1e and 1f, only monosubstituted product 2b could be obtained, because bromination can only occur at the para-position of the methoxyl group in 2b. For 2 -OMe substituted substrates 1a-1d, dibromination products were formed, because there are two para-positions of the two methoxyl groups. For 3 -OMe substituted substrates 1g and 1h, tribromination products were yielded, because there are three para-positions of the three methoxyl groups. As the -OMe is both ortho- and para- directing group, the excellent para-selectivity for bromination in this reaction system is very special and important, which is in line with the results reported in the literature (Synlett 2018, 29, 2275-2278, DOI: 10.1055/s-0037-1610980). Although it is not clear why the bromination is only para-selective (perhaps because there are some interactions between TFE and the substrate), the authors should mention and discuss this excellent para-selectivity for bromination in the manuscript as this is very important. The reason that "The electronic property of aromatic substrate" given by the authors is too general.

Other concerns have all been resolved.

Response: We thank Reviewer 1 for his/her kind assistance to review our response to Reviewer 2.

We agree with Reviewer 1's comments on the *para*-position selectivity of bromination reaction. As the Reviewer 1 said, owing to the strong directing effect of -OMe, the bromination has an excellent selective bromination occurring at para position of -OMe. Moreover, in reported bromination reaction of arenes, for the arenes with electron-donating group, the regioselectivity was observed in favor of the para position than ortho position (*Mol. Catal.* 2024, 553, 113777; *J. Org. Chem.* 2018, 83, 930–938; also *Synlett* 2018, 29, 2275-2278). In addition,

the degree of bromination (mono-, di-, and tri-bromination) could be also affected by the electronic property of aromatic rings (bromination failed with electron-deficient substrates). For more accurate description of this process, we revised the description for the depolymerization and bromination reaction in the revised manuscript as “*Notably, the result of bromination reactions for substrates with S-, G-, and H-type shows that the directing effect of methoxy group play an essential role for selective bromination, yielding brominated products with excellent para-selectivity (Fig. 3). Hence, the electronic property of substrates with S-, G-, and H-type directly affects the degree of overbromination and the directing effect of methoxyl group controls the site selectivity of bromination.*” We hope this discuss would exhibit more clear description about the bromination process.

Reviewer #3 (Remarks to the Author):

1. It is helpful to add the part about the effect of TFE for promotion the reactivity of NBS, some control experiments conducted, and the revised mechanism for this depolymerization and halogenation reaction.

Response: We thank you for your nice suggestion for exploration of TFE effect in the depolymerization and halogenation reaction.

2. The scale-up experiment is very challenging.

Response: We thank you for your advice about the scale-up experiment.

3. The mechanism is more clearly with the revised clarification in the legend.

Response: We appreciate you for positive comments on mechanism study and kind support for the publication of our manuscript.